# Rhein Improves Renal Fibrosis by Restoring Cpt1a-Mediated Fatty Acid Oxidation through SirT1/STAT3/twist1 Pathway

**DOI:** 10.3390/molecules27072344

**Published:** 2022-04-06

**Authors:** Xianrui Song, Zesen Du, Zhenqi Yao, Xiaoyan Tang, Mian Zhang

**Affiliations:** School of Traditional Chinese Medicine, China Pharmaceutical University, Nanjing 211198, China; 18364167506@163.com (X.S.); dzs951209@163.com (Z.D.); 3219020385@stu.cpu.edu.cn (Z.Y.); 3319020605@stu.cpu.edu.cn (X.T.)

**Keywords:** renal fibrosis, rhein, Cpt1a-mediated fatty acid oxidation, Twist1, epithelial–mesenchymal transition, SirT1/STAT3 pathway

## Abstract

The latest progress in the field of renal fibrosis mainly focuses on the new concept of “partial epithelial-mesenchymal transition (pEMT)” to explain the contribution of renal tubular epithelial (RTE) cells to renal fibrosis and the crucial role of fatty acid oxidation (FAO) dysfunction in RTE cells for the development of renal fibrosis. FAO depression is considered to be secondary or occur simultaneously with pEMT. We explored the relationship between pEMT and FAO and the effect of rhein on them. Intragastric administration of rhein significantly improved the levels of BUN, Scr, α-SMA, collagen 1A and histopathological changes in UUO-rats. Transcriptomic and metabolomic analyses revealed that abnormal signaling pathways were involved in EMT and FAO disorders. RTE cell experiments showed that TGF-β could inhibit the activity of Cpt1a, resulting in ATP depletion and lipid deposition. Cpt1a inhibitor induced EMT, while Cpt1 substrate or rhein inhibited EMT, indicating that Cpt1a-mediated FAO dysfunction is essential for RTE cells EMT. Further studies showed that Cpt1a activity were regulated by SirT1/STAT3/Twist1 pathway. Rhein inhibits RTE cell EMT by promoting Cpt1a-mediated FAO through the SirT1/STAT3/Twist1 pathway. Surprisingly and importantly, our experiments showed that FAO depression occurs before EMT, and EMT is one of the results of FAO depression.

## 1. Introduction

Fibrosis is essentially a repair response after tissue injury to protect the relative integrity of tissues and organs. However, if this repair reaction is persistent and out of control, it will cause excessive accumulation and deposition of extracellular matrix and lead to organ fibrosis [1,2]. Renal fibrosis is involved in the occurrence and development of almost all types of chronic kidney diseases (CKD), and it is also the main pathological pathway for all types of CKD to develop into end-stage renal failure. Renal fibrosis is characterized by glomerulosclerosis and tubulointerstitial fibrosis. Clinical studies have found that tubulointerstitial injury is more important than glomerular disease [3,4]. Tubulointerstitial fibrosis and tubular atrophy are the key factors determining the procession of progressive renal failure and the ten-year survival rate of kidney [5]. In addition, it has been shown that renal tubular epithelial (RTE) cells may be the initiator of various renal injury reactions [4,6]. Therefore, tubular interstitial fibrosis, especially tubular epithelial–mesenchymal transition (EMT), has become one of the research hotspots of CKD in recent years [7,8]. EMT means that epithelial cells lose cell adhesion and polarity to obtain the phenotypic characteristics of stromal cells. More and more evidence showed that EMT plays an important role in the process of renal interstitial fibrosis, and the degree of EMT was parallel to the severity of renal fibrosis [4,9]. Among the many regulatory factors of EMT, TGF-β is the most important upstream inducer, and TGF-β/Smad is the most important signal pathway to promote EMT. However, TGF-β is a cell regulator with extensive and complex biological effects. Direct inhibition of TGF-β may lead to serious negative effects. Therefore, factors or pathways associated with innate immunity and metabolism seem to increasingly become alternative targets for the development of new therapies for renal fibrosis. Up to now, there is still no effective therapy for renal fibrosis except hemodialysis and renal transplantation.

In recent years, it has been reported that the dysfunction of fatty acid oxidation (FAO) in RTE cells plays a key role in the development of renal fibrosis [10,11]. Proximal RTE cells have abundant mitochondria and high levels of baseline energy consumption. FAO is the preferred energy source for high metabolized RTE cells because it produces more ATP than glucose oxidation. Carnitine palmitoyl transferase 1a (Cpt1a) is the carrier of medium and long-chain fatty acids into mitochondria and the key rate-limiting enzyme for FAO. Overexpression of Cpt1a in RTE cells can protect mice from renal fibrosis by restoring oxidative metabolism and mitochondrial number [11]. Targeting depressed fatty acid metabolism may be a potential strategy to prevent and halt the development of renal fibrosis. 

Rhubarb (Rhei Radix et Rhizoma) is a commonly used traditional Chinese medicine with purgative effects. Pharmacological studies show that rhubarb has many actions, such as antibacterial, anti-inflammatory and anti-tumor activity, improving renal function, reducing blood lipid, weight loss and so on [12]. Rhein is one of the active anthraquinones isolated from rhubarb. It has the pharmacological effects of anti-tumor, anti-inflammation, improving diabetic nephropathy and antivirus [12]. It has also been reported that rhein can inhibit the process of renal fibrosis by regulating TGF-β, SIRT3 or STAT3 signals, apoptosis or autophagy of renal tubular cells [13,14,15,16]. However, the mechanism of rhein improving renal fibrosis is still very simple and insufficient. 

To find the main cellular pathway of rhein inhibiting renal fibrosis, we performed genome-wide transcriptome analysis of kidney samples from normal rats, unilateral ureteral obstruction (UUO) rats and rhein intervention rats. The results showed that rhein significantly reduced the levels of genes related to fibrosis and fatty acid metabolism. Thus, our study aimed to explore the role of rhein in mediating FAO in kidney fibrosis and the potential associated signaling mechanism.

## 2. Results

### 2.1. Rhein Attenuates UUO-Induced Renal Fibrosis in Rats

Firstly, we evaluated the effect of rhein (Figure 1A) on UUO-induced renal fibrosis in rats. After 14 days of UUO operation, the renal coefficient, blood urea nitrogen (BUN) and serum creatinine (Scr) increased significantly in rats of the UUO group (Figure 1B–D). Hematoxylin and eosin (H&E) staining and Masson’s staining showed that UUO operation caused obvious renal fibrosis, which was characterized by renal tubular atrophy, vacuolization, interstitial inflammation and extensive extracellular matrix (ECM) deposition in renal medulla and cortex (Figure 1E). IF staining also confirmed that UUO operation increased the protein levels of α-SMA and Col1A (Figure 1F). Compared with the UUO group, rhein could significantly reduce the renal coefficient and levels of BUN, Scr, α-SMA and Col1A, and effectively decrease the loss of parenchyma, tubular atrophy and ECM deposition in renal medulla and cortex (Figure 1B–F). These results demonstrated that rhein had a protective effect on UUO-induced renal fibrosis in rats.

### 2.2. UUO-Induced Renal Fibrosis in Rats Is Associated with FAO Dysfunction 

To explore the main mechanism of rhein improving renal fibrosis, we examined genome-wide transcript-level changes in the kidneys of rats in sham group, UUO group and rhein group. KEGG analysis of transcriptome data showed that the significant differentially expressed genes (q-value < 0.05) between sham group and UUO group mainly focused on inflammation, fibrosis and metabolic pathways (Appendix A). Inflammation is the initiating factor of fibrosis and is also closely associated with the development of fibrosis [17]. The differentially expressed signaling pathways related to inflammation and fibrosis mainly include PI3K-Akt, TNF, MAPK, NF-_k_B and Notch pathways (Figure 2A). Among them, Notch signaling pathway is the classical pathway that activates EMT, and PI3K-Akt and MAPK pathways are also related to EMT [18]. 

RNA sequencing analysis showed that the levels of fibrosis-related genes such as TGF-β, Acta2, vimentin, collagens, Twist1 and Fn1 increased significantly in UUO rats, and rhein could reduce the expressions of these genes (Figure 2B). This result was confirmed by detecting the mRNA expressions of Col1a1, Col3a1, Col4a1, TGF-β, vimentin, Acta2, Fn and Twist1 in the kidneys of rats in sham, UUO and rhein groups (Figure 2D). The high expressions of TGF-β, Twist1 (transcription factor inducing EMT) and vimentin (fibroblast marker) in UUO rats further suggested that EMT may be one of the important mechanisms of UUO-induced renal fibrosis. Rhein may improve renal fibrosis by inhibiting UUO-induced EMT.

Surprisingly, the pathway enrichment of transcriptomic data also showed a strong signal that energy metabolism was markedly down-regulated in UUO-induced rats (Figure 2A). Although cAMP signaling pathway was also strongly enriched in UUO-induced rats, many studies have found that EMT in renal fibrosis is closely associated with cAMP signaling pathway [19,20]. On the other hand, rhein has no obvious callback effect on the genes associated with cAMP pathway, especially the key genes such as Adcy 1 (adenylyl cyclase 1), Pde4 (phosphodiesterase 4) and Creb3l3 (cAMP response element binding 3l3) (Appendix A). Among these enriched metabolism pathways, fatty acid degradation and amino acid metabolism (valine, leucine and isoleucine degradation and glycine, serine and threonine metabolism) are all related to FAO (Figure 2A). Both RNA sequencing and mRNA detection displayed that the levels of FAO-related genes (Cpt1, Cpt2, Acot1, Decr1, Echs1, etc.) in UUO rats were markedly lower than those in sham rats, but rhein treatment could more or less reverse the low expression of these genes (Figure 2C,E). This result demonstrated that UUO operation inhibited fatty acid degradation by downregulating the levels of FAO-related genes in the rat kidney. Rhein may upregulate the expressions of these genes to promote fatty acid degradation. 

Although it has been reported that both tubular EMT and FAO dysfunction play important roles in the development of renal fibrosis [7,8,9,10,11], the relationship between them is still unclear. To understand whether renal fibrosis, especially EMT, was related to FAO, untargeted mass spectrometry metabolomics was performed on the kidneys from rats in sham, UUO and rhein groups. With VIP > 1.0 and *p* < 0.05, 34 differential metabolites between the UUO group and sham group were identified (Figure 2F, Appendix A), and rhein could significantly recover the levels of 22 differential metabolites (Figure 2F, metabolites in red). Further analysis using HMDB showed that 16 of the 22 metabolites were related to lipid metabolism. KEGG enrichment also showed that the differential metabolites were mainly involved in pathways related to lipid metabolism such as glyoxylate and dicarboxylate metabolism and glycerophospholipid metabolism (Figure 2G). These data suggested that UUO-induced fibrosis was closely associated with the changes of lipid metabolism, and rhein alleviated renal fibrosis mainly by normalizing abnormal lipid metabolism. More detailed analysis showed that 22 differential metabolites were significantly correlated with Cpt1a (rate-limiting enzyme of FAO) gene and 13 with Twist1 (transcription factor inducing EMT) gene, of which 10 differential metabolites were correlated with both Cpt1a gene and Twist1 gene (Figure 2H and Appendix A). This result indicated that Twist1 induced EMT in renal fibrosis may be closely associated with FAO dysfunction. Therefore, the effect of fatty acid metabolism on EMT of renal tubular epithelial (RTE) cells was investigated in vitro.

### 2.3. Cpt1-Mediated FAO Depression Is Necessary for TGF-β-Induced EMT in RTE Cells

CCK-8 assay showed that rhein at a concentration of 0.1–50 μmol/L had no significant effect on the viability of RTE cells (Figure 3A). To assess the effect of rhein on EMT, RTE cells were stimulated with TGF-β (10 ng/mL) and simultaneously treated with different concentrations of rhein for 24 h, and the protein expressions of E-cadherin, α-SMA and Col1A were examined. Compared with the control group (Figure 3B), TGF-β successfully induced EMT in RTE cells, which was marked by decreased expression of E-cadherin (*p* < 0.001) and increased expressions of α-SMA (*p* < 0.01) and Col1A (*p* < 0.001). Rhein (50 μmol/L) effectively inhibited TGF-β-induced EMT by increasing the expression of E-cadherin (*p* < 0.01) and decreasing the expressions of α-SMA (*p* < 0.001) and Col1A (*p* < 0.001) (Figure 3B). BODIPY staining showed that the lipid droplets in TGF-β-induced RTE cells obviously increased compared to the blank (Figure 3C), accompanied by a significant decrease in intracellular ATP level (Figure 3D). It was suggested that fatty acid degradation was blocked in TGF-β-induced cells. However, rhein could reverse the effect of TGF-β on RTE cells (Figure 3C,D). Cpt1 is a key rate-limiting enzyme for the oxidation of medium and long-chain fatty acids. The Cpt1 inhibitor etomoxir also significantly decreased the content of intracellular ATP in RTE cells (Figure 3D), indicating that FAO is the key contributor to intracellular ATP levels in RTE cells. As shown in Figure 3E,F, the expression and activity of Cpt1 in TGF-β-induced cells decreased significantly (*p* < 0.01, 0.001) compared to blank samples, which could be reversed by rhein. 

To investigate whether Cpt1 was associated with EMT, we detected the mRNA expressions of E-cadherin (epithelium indicator), vimentin (fibroblast indicator), Acta2 and Col1a1 in RTE cells induced by TGF-β or etomoxir. Compared with the blank group, both TGF-β and etomoxir decreased the expression of E-cadherin and increased the expressions of vimentin, Acta2 and Col1a1 (Figure 3G), together with a down-regulated Cpt1 activity (Figure 3H). Carnitine is the substrate of Cpt1. Carnitine could restore the Cpt1 activity downregulated by TGF-β (Figure 3H) to reverse TGF-β-induced EMT, that is, increasing the expression of E-cadherin and reducing the expressions of vimentin, Acta2 and Col1a1 in RTE cells (Figure 3I). These results suggested that Cpt1 is a key factor in TGF-β-induced EMT. Furthermore, TGF-β significantly downregulated the mRNA and protein expression of Cpt1a, but had no effect on that of Cpt1b and Cpt1c (Figure 3J,K). Rhein could increase the Cpt1 activity (Figure 3F) and the mRNA and protein expression of Cpt1a in TGF-β-induced RTE cells (Figure 3K). Collectively, TGF-β could induce EMT by blocking Cpt1a-mediated FAO, and rhein could reverse the effect of TGF-β by promoting the activity and expression of Cpt1a.

### 2.4. Twist1 Is Critical for Cpt1-Mediated FAO Dysfunction in RTE Cells 

Twist1 is a key transcription factor to induce EMT [21]. Compared to the blank cells, the mRNA and protein expression of Twist1 in TGF-β-induced RTE cells significantly upregulated, and rhein could reduce the high expressions induced by TGF-β (Figure 4A,B). To investigate the role of Twist1, RTE cells were transfected with Twist1-specific siRNA to silence endogenous Twist1. 

Compared with the model (TGF-β + Ctrl siRNA) group, both rhein and Twist1-transfection could significantly increase the mRNA and protein expression of E-cadherin and reduce the same expressions of vimentin, Acta2 and Col1a1 (Figure 4C,D). Moreover, the protein and mRNA expressions of E-cadherin, vimentin, α-SMA and Col1a1 in Twist1-transfected cells did not change whether TGF-β or rhein was added or not (Figure 4C,D), except for the mRNA of vimentin and Acta2. Further, to understand the relationship between Twist1 and Cpt1a, the expression and activity of Cpt1a in Twist1-transfected cells were detected. After transfection of Twist1, the mRNA expression of Cpt1a in RTE cells did not change whether induced by TGF-β or treated by rhein (Figure 4E). Compared with the blank group, TGF-β induction could decrease the activity and protein expression of Cpt1, increase lipid deposition and ATP depletion, while Twist1 transfection abolished the above effects of TGF-β (Figure 4F–I). These results suggested that Twist1 is essential for Cpt1a-mediated FAO dysfunction, which leads to EMT of RTE cells. Rhein promoted Cpt1a-mediated FAO by regulating Twist1 to prevent TGF-β-induced EMT in RTE cells. 

### 2.5. Rhein Blocks SirT1/STAT3 Signaling to Inhibit Twist1 Transcription in TGF-β-Induced RTE Cells

It was reported that TGF-β can activate STAT3 in RTE cells [22]. To assess whether STAT3 is required for TGF-β-induced Twist1 activation, we measured the protein expressions of STAT3 and Twist1 in TGF-β-induced cells with or without WP1066, an inhibitor of STAT3 activity. As shown in Figure 5A, TGF-β induced the high expressions of *p*-STAT3 and Twist1. Both rhein and WP1066 could downregulate the high expressions of *p*-STAT3 and Twist1 induced by TGF-β, thereby reversing TGF-β-induced EMT (manifested by the increase in E-cadherin and decrease in α-SMA and Col1A) (Figure 5A). Silencing STAT3 also significantly reduced the expression of Twist1 in TGF-β-induced cells (Figure 5B). Moreover, ChIP assay indicated that STAT3 bound directly to the Twist1 promoter to activate the transcription of Twist1 mRNA in TGF-β-induced cells, and rhein could block the binding of STAT3 to Twist1 promoter (Figure 5C). The above results exhibited that STAT3 was critical for TGF-β-induced Twist1 activation, and rhein could inhibit STAT3 phosphorylation. 

Sustained activation of STAT3 was a key molecular event in TGF-β-induced EMT. SirT1 is a deacetylase closely related to energy metabolism. It has been reported that there is a direct interaction between SirT1 and STAT3. There are four lysine acetylation sites near the SH2 domain of STAT3, and SirT1 inhibits the phosphorylation of STAT3 through deacetylation of these sites [23]. WB detection on SirT1 and STAT3 proteins in RTE cells showed that TGF-β reduced the expression of SirT1 and increased the phosphorylation and lysine acetylation of STAT3 (Figure 5D,E). However, rhein and resveratrol (activator of SirT1) reversed the effects of TGF-β on SirT1 and STAT3 (Figure 5D,E), and further reversed TGF-β-induced Twist1 expression and EMT-related indicators such as Col1A, α-SMA and E-cadherin (Figure 5F). SirT1 has a negative regulatory effect on STAT3. Therefore, after silencing SirT1 with specific siRNA, the phosphorylation and lysine acetylation of STAT3 increased significantly and remained unchanged, whether treated with TGF-β or rhein (Figure 5G,H). Together, these results confirmed that SirT1/STAT3 signaling was upstream of Twist1, and rhein could block this signaling to inhibit the transcription of Twist1 gene in TGF-β-induced RTE cells. 

### 2.6. Rhein Inhibits SirT1/STAT3/Twist1 Pathway to Promote Cpt1a-Mediated FAO in UUO-Induced Rats

To verify the above mechanism of rhein improving renal fibrosis, we detected the indicators related to EMT and FAO in UUO-induced rat kidneys. Compared with the sham rats, the protein expression of E-cadherin reduced significantly in UUO rats, together with obviously increased α-SMA and Col1A expressions (Figure 6A), indicating that EMT may be one of important pathological mechanisms of renal fibrosis. At the same time, UUO-operation significantly increased the lipid accumulation (Figure 6B), reduced the protein and mRNA expressions of Cpt1a (Figure 6C,D), increased the protein expression of Twist1 and the phosphorylation and acetylation of STAT3 (Figure 6C,E), and decreased the protein expression of SirT1 (Figure 6E). Both rhein and resveratrol (activator of SirT1) could reverse the effects of UUO-operation (Figure 6A–E). Collectively, rhein protects rats from renal fibrosis by regulating SirT1/STAT3/Twist1 pathway to promote Cpt1a-mediated fatty acid degradation. 

## 3. Discussion 

For traditional Chinese medicine, rhubarb is the most commonly used effective drug for the treatment of chronic renal failure [24]. Rhein, as one of the effective components of rhubarb, has also been reported to significantly improve renal fibrosis [16], which was confirmed by our animal experimental results (Figure 1). However, there were few studies on the mechanism of rhein against renal fibrosis. Here, we present that rhein upregulated Cpt1a-mediated FAO through the SirT1/STAT3/Twist1 pathway to improve renal fibrosis. 

Due to the central importance of myofibroblasts in the formation of extracellular matrix, previous studies on the pathogenesis of renal fibrosis mainly focused on the transdifferentiation of fibroblasts into myofibroblasts [10]. In recent years, increasing amounts of data revealed that RTE cells are not only the victims of renal injury, but also the key driving force of renal inflammation and renal fibrosis [7,25]. Renal tubules are the hub connecting glomerulus and renal interstitium. They play a key role in excreting metabolites and maintaining body fluid balance and acid–base balance. However, RTE cells are vulnerable to pathogenic factors, and the persistent injury of epithelial cells will lead to EMT. Unlike other epithelial cells, RTE cells undergo a “partial EMT” (pEMT); that is, dedifferentiated RTE cells acquire mesenchymal characteristics but still adhere to the basement membrane [26] and co-express epithelial (E-cadherin) and mesenchymal (vimentin) markers [27]. This is consistent with our experimental results (Figure 3G,I). Recent studies showed that a pEMT is sufficient to induce RTE cell dysfunction by triggering cell cycle arrest, metabolic alternation and inflammation, resulting in a disturbed epithelial-mesenchymal crosstalk to lead to renal fibrosis [9]. Therefore, pEMT of RTE cells may be a new target for the prevention and treatment of renal fibrosis. Our in vitro experiments showed that rhein could effectively inhibit the pEMT of RTE cells, which may be a potential therapeutic agent for renal fibrosis. 

RTE cells have a high energy demand and primarily rely on FAO as their energy source [28]. Many studies have shown that enzymes and key regulators of FAO were reduced in fibrotic kidneys, and dysfunctional FAO results in accumulation of intracellular lipids and the acceleration of interstitial fibrosis [10,29]. A more detailed study showed that dysfunctional FAO, rather than intracellular lipid accumulation, induces the development of renal fibrosis [11]. In this study, we found that the levels of FAO-related genes reduced significantly in UUO rats (Figure 2C,E). Cpt1a is the rate limiting enzyme of FAO. Our metabolomic study demonstrated that 22 of the 34 differential metabolites between UUO and sham rats were significantly associated with Cpt1a gene (Appendix A), indicating that Cpt1a-mediated FAO plays a very important role in UUO-induced renal fibrosis. Further experiments showed that Cpt1 inhibitor could induce EMT in RTE cells, while Cpt1 substrate could restore Cpt1 activity and reverse TGF-β-induced EMT (Figure 3G–I). It was reported that overexpression of Cpt1a can protect mice from renal fibrosis [11]. Consistent with the report, the above results suggested that Cpt1 is crucial for TGF-β-induced EMT, and rhein can restore the mRNA/protein expression and activity of Cpt1 downregulated by TGF-β. Of course, further research is needed to explore how Cpt1-mediated FAO depression leads to EMT of RTE cells.

Twist proteins exhibit bi-functional roles as both activators and repressors of gene transcription [30]. Twist1 is a critical regulator of EMT, but how it regulates EMT is not clear. Our metabolomic study showed that 13 of the 34 differential metabolites between UUO and sham rats were significantly associated with Twist1 gene (Appendix A). Further, we found that after silencing Twist1, TGF-β could neither downregulate Cpt1a-mediated FAO nor induce EMT (Figure 4). This suggested that Twist1 is upstream of Cpt1a and a key factor in Cpt1a-mediated FAO depression of RTE cells. It has been considered that FAO depression may occur secondary or in parallel to EMT [18]. However, our results showed that Twist1 leads to FAO dysfunction by inhibiting the transcription of Cpt1a gene, and then resulting in EMT of RTE cells. Rhein could inhibit the expression and activity of Twist1 to promote Cpt1a expression. Whether Cpt1a reverses renal fibrosis through its interaction with Twist1 remains to be further studied. 

STAT3 mediates TGF-β1-induced Twist1 expression leading to prostate cancer invasion [31]. In this study, we found that both STAT3 inhibitor and silencing STAT3 could downregulate TGF-β-induced high expression of Twist1 in RTE cells. Moreover, ChIP experiment demonstrated that STAT3 activates the transcription of Twist1 mRNA by directly binding to the Twist1 promoter (Figure 5C). These results suggested that STAT3 is an upstream regulator of Twist1 and is required for Twist1 expression. SirT1 is a NAD^+^-dependent deacetylase. There is an inverse relationship between SirT1 and the TGF-β signaling pathway in renal fibrosis, and SirT1 has the effect of counteracting renal fibrosis [32]. STAT3 activity is inhibited by SirT1-mediated deacetylation in hepatocytes [23]. In this study, we found that TGF-β decreased the expression of SirT1 and increased the activation of STAT3 in RTE cells, and both rhein and SirT1 activator could reverse the effect of TGF-β (Figure 5D,E). Moreover, silencing SirT1 in RTE cells resulted in a significant increase in the lysine acetylation and phosphorylation of STAT3 (Figure 5G,H), indicating that STAT3 is regulated by SirT1-mediated deacetylation. Therefore, our in vitro and in vivo studies have demonstrated that Twist1 transcription is adjusted by SirT1/STAT3 pathway, and Twist1 is essential for Cpt1a-mediated FAO dysfunction. Rhein could improve renal fibrosis, at least in part, by regulating this pathway. Twist1 negatively adjusts Cpt1a. Cpt1a inhibition causes FAO dysfunction, which leads to EMT of RTE cells. According to the hypothesis of “pEMT” [9], the pEMT cells lead to a disordered crosstalk with mesenchymal cells, which will lead to tubular interstitial fibrosis (Figure 7). 

## 4. Materials and Methods

### 4.1. Reagents

Rhein (purity ≥ 98%, 113443-70-1) and resveratrol (purity ≥ 98%, 501-36-0) were purchased from Chengdu Biopurify Phytochemicals Ltd. (Chengdu, China) and Chengdu DeSiTe Biological Technology Co, Ltd. (Chengdu, China). Carnitine (541-15-1), etomoxir (124083-20-1), TGF-β (HY-P7118) and WP1066 (857064-38-1) were purchased from MedChemExpress biological engineering Co, Ltd. (CA, USA). 

*Test kits* BUN (C013-2-1) and Scr assay kits (C011-2-1) were purchased from Nanjing Jiancheng Bioengineering Institute (Nanjing, China). Type IV collagenase (C5138) were obtained from Sigma-Aldrich (St Louis, MO, USA). Hoechst dye (C1011) and ATP assay kit (S0026) were obtained from Beyotime Institute of Biotechnology (Shanghai, China). Hematoxylin-Eosin (H&E) (G1120) staining kit and Masson’s trichrome stain kit (G1340) were purchased from Solarbio Life Sciences (Beijing, China). CCK-8 assay kit (CK04) was purchased from Dojindo Laboratories (Fukuoka, Japan). Cpt1 activity assay kit (ml076617) was obtained from Shanghai Mlbio Biotechnology Co, Ltd. (Shanghai, China). BODIPY™ 493/503 (4,4-difluoro-1,3,5,7,8-pentamethyl-4-bora-3a,4a-diaza-s-indacene) dye (D3922) was purchased from Thermo Fisher Scientific (Waltham, MA, USA). 

*Transfection reagents* Twist1-specific siRNA (AM16708), Lipofectamine 2000 (11668-019) were purchased from Thermo Fisher Scientific (Waltham, MA, USA). siRNA transfection reagent (sc-29528), SirT1 siRNA (sc-108043), STAT3 siRNA (sc-270027), control siRNA (sc-37007) was purchased from Santa Cruz Biotechnology (Santa Cruz, CA, USA). 

*PCR and WB reagents* PrimeScript^TM^ RT reagent Kit (R122-01), SYBR^®^ Green PCR Core Reagents kit (Q131-02) were purchased from Vazyme Biotech Co, Ltd. (Shanghai, China). Dulbecco’s modified eagle’s medium (DMEM) (31600), fetal bovine serum (FBS) (10270-106), Opti-MEM Reduced-Serum Medium (31985-070) were obtained from Gibco (Grand Island, NY, USA). ECL enhanced chemiluminescence (36222ES76) was purchased from Yeasen Biotech Co, Ltd. (Shanghai, China). Bicinchoninic Acid (BCA) protein assay (P0012), DAPI dye (C1005), protein A+G agarose beads (P2012) were obtained from Beyotime Institute of Biotechnology (Shanghai, China). Chromatin-Immunoprecipitation (ChIP) kit (#9002) were purchased from Cell Signaling Technology (Beverly, CA, USA). 

*Antibodies* Rabbit anti-β-actin antibody (AP0060) was purchased from Bioworld Technology Inc (Nanjing, China). Mouse anti-Cpt1a antibody (sc-393070) were purchased from Santa Cruz Biotechnology (Santa Cruz, CA, USA). Rabbit anti-vimentin antibody (10366-1-AP), rabbit anti-E-cadherin antibody (20874-1-AP) and rabbit anti-Twist1 antibody (25465-1-AP) were purchased from Proteintech Group (Wuhan, China). Rabbit anti-α-SMA antibody (#19245), mouse anti-Col1A antibody (#66948), rabbit anti-Cpt1a antibody (#97361), anti-STAT3 rabbit polyclonal antibody (#12640), rabbit anti-*p*-STAT3 antibody (#9145), rabbit anti-acetylated-Lysine (#9441), rabbit anti-SirT1 antibody (#2496), FITC-goat anti-rabbit IgG (#4414) and Alexa-goat anti-mouse IgG (#4412) were purchased from Cell Signaling Technology (Beverly, CA, USA). 

### 4.2. Unilateral Ureteral Obstruction (UUO)-Induced Renal Fibrosis in Rats

Male Wistar rats (170–190 g, 7–8 weeks old) were purchased from Shanghai Sippe-Bk Lab Animal Co Ltd. (SCXK(Hu) 2018–0006). The rats were placed in colony cages and had free access to food and water. The light/dark cycle was 12 h and the temperature was maintained at 21 ± 2 °C. Animal welfare and experimental procedures complied with the Provisions and General Recommendations of the Chinese Experimental Animals Administration Legislation. The animal experiment protocol was approved by the Animal Care and Use Committee of China Pharmaceutical University (permit No. SCXK (Hu) 2018–0006). The animals were acclimated to the laboratory for at least 7 days before use in the experiments. 

Renal fibrosis model was established by unilateral ureteral obstruction (UUO) in rats. The rats were randomly divided into a sham group, model group and drug groups. Under isoflurane (3–4%, 0.6–0.8 L/min) inhalation anesthesia, the left ureter of rat was ligated in two places with 4/0-gauge surgical silk and cut between the ligatures, except for the sham operation group. Rhein (100 mg/kg) or resveratrol (20 mg/kg) were dissolved with 0.5% sodium carboxymethylcellulose (CMC-Na) and given to rats by intragastric administration once a day from days 1 to 14 after UUO surgery. The rats in the sham and model groups received an equivalent volume of 0.5% CMC-Na. The animal body weights were measured every two days. On day 15, the rats were sacrificed under isoflurane inhalation anesthesia. The blood was collected from the abdominal aorta, and then the kidneys were removed and weighed. The renal coefficient was calculated as kidney weight (mg)/body weight (g) × 100%. 

### 4.3. Urea Nitrogen (BUN) and Serum Creatinine (Scr) Analysis

The collected blood was centrifuged at 3000 rpm for 15 min and the supernatant (serum) was separated. According to the manufacturer’s instructions, the contents of BUN and Scr in serum were determined by corresponding commercial kits. 

### 4.4. Histopathological and Immunofluorescence (IF) Examination

The removed kidneys were fixed in 4% paraformaldehyde, embedded in paraffin and cut into 5-μm-thick sections. For histopathological assay, the sections were stained with hematoxylin-eosin (H&E) to observe pathological changes or Masson’s trichrome to evaluate collagen deposition. Renal pathological changes and collagen deposition were estimated by professionals using morphometric methods [33]. For IF assay of renal tissue, the sections were blocked with normal goat serum for 30 min, and then stained with primary antibodies of α-SMA, Col1A, or a mixture of Cpt1a and Twist1, respectively. The sections were incubated with secondary antibodies for 2 h and counterstained with DAPI for 5 min at room temperature. For the IF assay of cells, RTE cells were inoculated on 24-well plates (2 × 10^5^ cells/well) and starved for 12 h. Then the cells were transfected with Twist1 siRNA for 6 h, stimulated with TGF-β (10 ng/mL) for 24 h and washed with PBS. After blocking, the cells were stained with Cpt1a antibody, incubated with secondary antibody and counterstained with DAPI. All sections were inspected using an Olympus microscope (Tokyo, Japan).

### 4.5. Transcriptomics Analysis

Total RNA from rat kidney was extracted using RNeasy Plus kit with genomic DNA removal step. The concentration and quality of RNA were assessed by NanoDrop ND-1000 spectrophotometer. cDNA library construction and sequencing were performed by Novogene Bio Technology Co Ltd. (Beijing, China). 

The raw sequencing reads were cleaned by removing adaptors and low-quality reads (Q-value < 20). HISAT2 software was used to map the clean reads to the reference genome. Gene expression was quantified by calculating fragments per kilobase of exon per million fragments mapped (FPKM) values using the StringTie software package. Common and significantly changed genes were selected to determine whether there were significant differences in gene expression between sham group and UUO group or between UUO group and rhein group. Significantly different expression between groups was determined according to the following criteria: q-value < 0.05 and log_2_(Fold Change) > 1.

### 4.6. Metabolomics Analysis

Non-targeted metabolomics were performed on rat kidney samples. The renal extraction was performed according to the methods previously described [34]. Briefly, the kidney tissue was homogenized with 9 times water and centrifuged. Each 100 μL of supernatant was extracted with 300 μL of acetonitrile. The extract was dried with nitrogen, dissolved with 200 μL of 50% acetonitrile solution, and centrifuged. The supernatant was analyzed by liquid chromatography-mass spectrometry (Agilent Technologies 6530 Accurate-Mass Q-TOF system, Santa Clara, CA, USA). 

Supervised orthogonal partial least-squares discriminant analysis (OPLS-DA) was applied to identify the differences of metabolic phenotypes among groups. The features with variable importance in projection (VIP) scores higher than 1 in the OPLS-DA model and *p*-values lower than 0.05 using *t*-test corrected by False Discovery rate were selected, and their corresponding metabolites were identified. For identification of metabolites in TOF-MS analysis, the Human Metabolome Database (HMDB) was used. 

### 4.7. Cell Preparation and Culture

The primary rat renal tubular epithelial (RTE) cells were isolated from the renal tissues of 3-7-day-old Wistar rats according to the procedure described in the literature [35]. Briefly, the renal cortex is cut into small pieces and digested with DMEM containing 0.1% Type IV collagenase for 30 min. After washing, filtration and centrifugation, the pellets were resuspended in complete medium and seeded in culture flask for 30 min. The supernatant medium was transferred to another cell flask to remove fibroblasts and other miscellaneous cells. After growing to 80% confluence, the cells could be used for experiments. For the experiment, RTE cells were starved for 12 h and then cocultured with TGF-β (10 ng/mL, model) and drugs (rhein, inhibitor or activator) for 24 h for the detection of various indexes.

### 4.8. Cell Viability Assay 

Cell viability was determined using the CCK-8 assay kit. RTE cells were seeded in 96-well plates (1 × 10^4^ cells/well) overnight. After starvation of 12 h, RTE cells were incubated with 0.1, 1, 10, 50 and 100 μmol/L of rhein for 24 h, and then treated with CCK-8 solution for 1 h. Cell viability was expressed as absorbance value which was measured at 450 nm using a microplate reader (Thermo Varioskan LUX, Waltham, MA, USA).

### 4.9. Western Blot (WB) and Immunoprecipitation (IP)

For Western blot, the proteins of renal tissue or RTE cells were extracted with RIPA lysis buffer. The lysates were centrifuged, and the supernatants were collected. The protein contents were measured by Pierce Bicinchoninic Acid (BCA) protein assay. The proteins were dissolved with 10–12% SDS-PAGE and transferred to PVDF membrane. After blocking, the proteins were first incubated with primary antibodies, then incubated with corresponding secondary antibodies, and finally visualized with ECL enhanced chemiluminescence reagent.

For immunoprecipitation, the collected supernatant of the protein lysate was incubated with STAT3 antibody overnight and mixed with protein A + G agarose beads. The mixture was incubated under rotary stirring for 4 h and centrifuged. The precipitates were washed and boiled in loading buffer for 15 min and analyzed by Western blot. 

### 4.10. q-PCR and Chromatin Immunoprecipitation Assay (ChIP)

Total RNA was extracted from RTE cells or renal tissue using Trizol reagent (Invitrogen, Carlsbad, CA, USA) according to the manufacturer’s directions. The quality and quantity of extracted RNA were determined by NanoDrop 2000 spectrophotometer (Thermo Fisher Scientific). The extracted total RNA was reverse transcribed into cDNA using the PrimeScriptTM RT reagent Kit with gDNA Eraser. qPCR was performed to analyze the gene expression profiles using Stratagene Mx3000P system (Agilent Technologies) and SYBR^®^ Green PCR Core Reagents kit. The measurements were standardized to the expression of β-actin gene. The 2^−ΔΔCt^ method was used for relative quantification of target genes. Genes and primer sequences were listed in Appendix A.

ChIP assay was performed using an Imprint Chromatin-Immunoprecipitation kit according to the manufacturer’s instructions. Chromatin prepared from RTE cells was immunoprecipitated using anti-STAT3 antibody or IgG. The primer sequences for a Twist1 promoter are shown below. Forward: 5′-TCCTTCCAGGTTGTTCAGCG-3′, reverse: 5′-AGACACCGGATCTATTTGCATTT-3′. The STAT3 binding site sequence of Twist1 may be 5′-TTTCTTGGAAA-3′. The relative amount of precipitated DNA was quantified by real-time PCR and normalized to average values of input DNA. 

### 4.11. RTE Cell Transfection

RTE cells were inoculated on 12-well plates and transfected separately with Twist1-specific siRNA, SirT1 siRNA or STAT3 siRNA according to the manufacturer’s instructions. After transfection, the cells were cultured in fresh medium for 48 h and then exposed to TGF-β (10 ng/mL) with or without rhein (50 μmol/L) for 24 h. The cells were collected for further examination.

### 4.12. Determination of ATP Activity and Cpt1 Activity

For the determination of ATP activity, RTE cells transfected with Twist1 siRNA were grown in 96-well plates (1 × 10^4^ cells/well) and starved for 12 h, and then exposed to TGF-β (10 ng/mL) or etomoxir (50 μmol/L) for 24 h. ATP activity was measured by detecting chemiluminescence with a microplate reader (Thermo Varioskan LUX, Waltham, MA, USA) according to the instructions of the ATP assay kit manufacturer.

For the determination of Cpt1 activity, RTE cells transfected with Twist1 siRNA, the proteins of RTE cells or renal tissues were extracted with lysis reagent. Cpt1 activity was measured by visible spectrophotometry according to the instructions of manufacturer of Cpt1 activity assay kit.

### 4.13. BODIPY Staining 

Lipid droplets were stained with BODIPY™ 493/503 dye. Briefly, RTE cells or renal tissues sections to be tested were incubated with dye for 30 min and staining with Hoechst in the dark for 10 min. The stained lipid droplets (with green fluorescence) were observed and photographed by confocal scanning microscope (Zeiss LSM 700, Jena, Germany) at excitation wavelength 488 nm and emission wavelength 525 nm. 

### 4.14. Statistical Analysis

The biological data were expressed as the mean ± SD (standard deviation) of at least five independent experiments unless otherwise indicated. Statistical analysis was performed using GraphPad Prism 7.0 software. *t*-test and Tukey test were used to compare the statistical significance of the difference between the sample and the control. The value of *p* < 0.05 was considered statistically significant.

Metabolic pathways associated with identified genes or metabolites were established by integrating the Kyoto Encyclopedia of Genes and Genomes (KEGG). Pearson correlation coefficient was used to analyze the correlation between differential metabolites and target genes, and r-3.3.3 method was used for other correlation analysis.

## 5. Conclusions

In this study, we demonstrated that rhein prevents renal fibrosis by promoting Cpt1a-mediated FAO through SirT1/STAT3/Twist1 pathway. In renal fibrosis, Twist1 is a driving force of EMT in RTE cells by negatively regulating Cpt1a-mediated FAO. Different from the reported views, our experiment showed that FAO depression occurs before EMT, and EMT is one of the results of FAO depression. STAT3 enhances Twist1 gene transcription by directly binding to its promotor, and STAT3 is negatively adjusted by SirT1. Therefore, SirT1, STAT3, Twist1 and Cpt1a may represent useful targets for the prevention of renal fibrosis. Rhein can improve renal fibrosis by blocking SirT1/STAS3/Twist1/Cpt1a-depandent FAO dysfunction, which may also be one of the mechanisms for rhubarb drugs. Rhein and rhubarb drugs are promising potential therapeutic agents for renal fibrosis. 

## Figures and Tables

**Figure 1 molecules-27-02344-f001:**
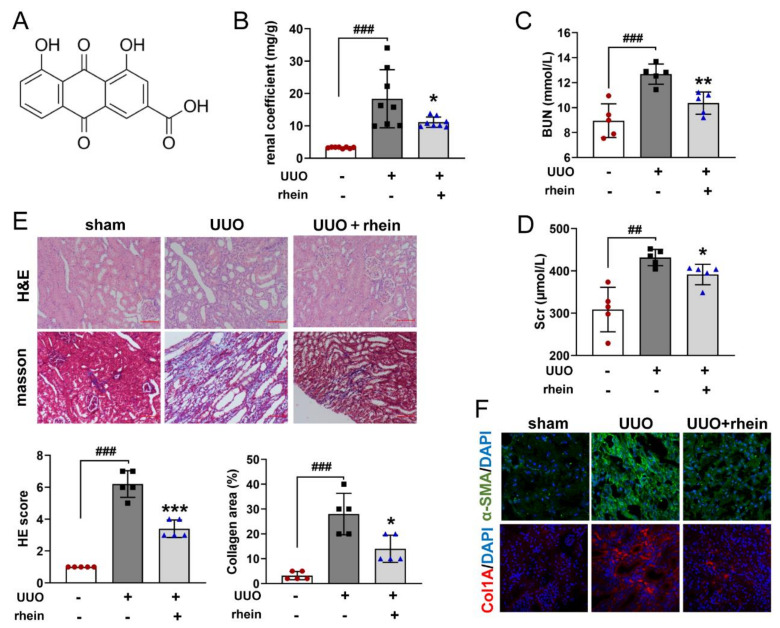
Rhein attenuates UUO-induced renal fibrosis in rats. The left ureter of rat was ligated in two places and cut between the two ligatures (UUO operation). The rats were i.g. administered with or without rhein at 100 mg/kg per day from days 1 to 14 after UUO operation. (**A**) Structure of rhein; (**B**) renal coefficient; (**C**,**D**) contents of BUN and Scr in serum; (**E**) representative images of kidney samples stained with H&E and Masson (scale bars, 100 μm), and quantitative analysis of HE score and collagen area percentage; (**F**) representative images of kidney samples immunostained with α-SMA and Col1A (*n* = 3). Data were expressed as mean ± SD (*n* = 5). ^##^ *p* < 0.01, ^###^ *p* < 0.001, vs. sham rats; * *p* < 0.05, ** *p* < 0.01, *** *p* < 0.001, vs. UUO rats.

**Figure 2 molecules-27-02344-f002:**
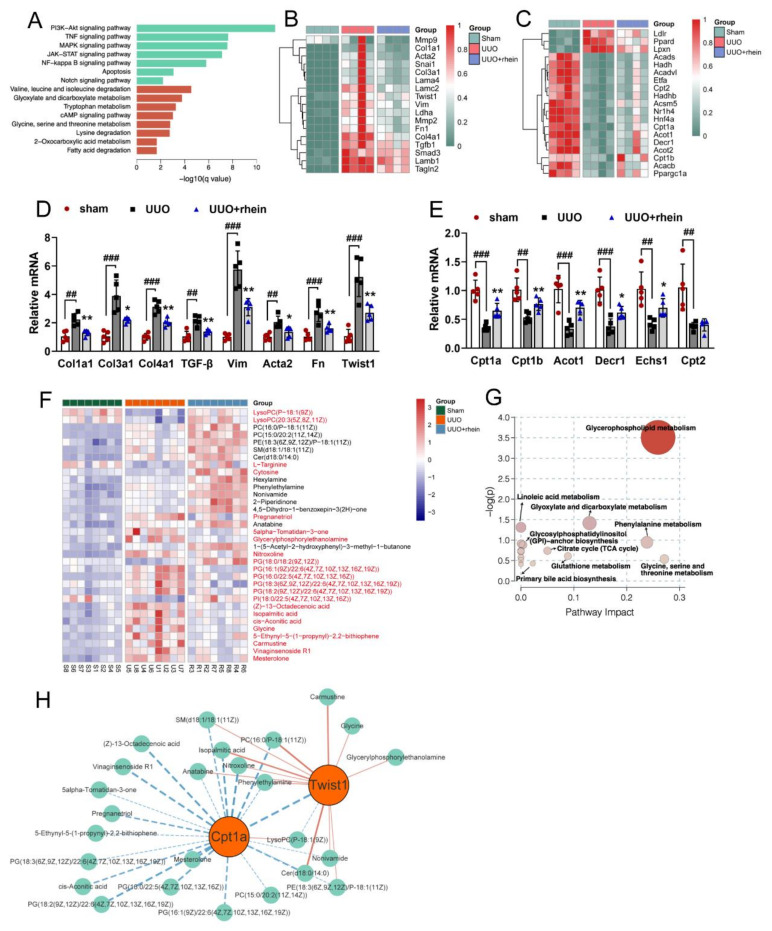
Transcriptomic and metabolomic analysis of UUO-induced rat kidneys. Rats were operated and treated in the same manner as in Figure 1. Total RNA extracted from kidney samples was used for transcriptomic analysis, and acetonitrile extract from kidney samples was used for metabolomic analysis. (**A**) KEGG enrichment pathways of differential genes between UUO and sham rats with q-value < 0.05 and log_2_(FoldChange) > 1; RNA sequencing analysis of genes associated with (**B**) fibrosis (*n* = 4) and (**C**) fatty acid metabolism (*n* = 4); mRNA expressions associated with (**D**) fibrosis (Col1a1, Col3a1,Col4a1, TGF-β, Vim, Acta2, Fn1 and Twist1) and (**E**) fatty acid metabolism (Cpt1a, Cpt1b, Acot1, Decr1, Echs1 and Cpt2) in rat kidney samples were determined (*n* = 5); (**F**) heatmap of metabolites in rat kidney samples (*n* = 8); (**G**) KEGG enrichment pathways of differential metabolites between UUO and sham rats with VIP > 1 and *p* < 0.05; (**H**) network diagram of Twist1 gene, Cpt1a gene and differential metabolites in kidney of UUO and sham rats. Data were expressed as mean ± SD. ^##^ *p* < 0.01, ^###^ *p* < 0.001, vs. sham rats; * *p* < 0.05, ** *p* < 0.01, vs. UUO rats.

**Figure 3 molecules-27-02344-f003:**
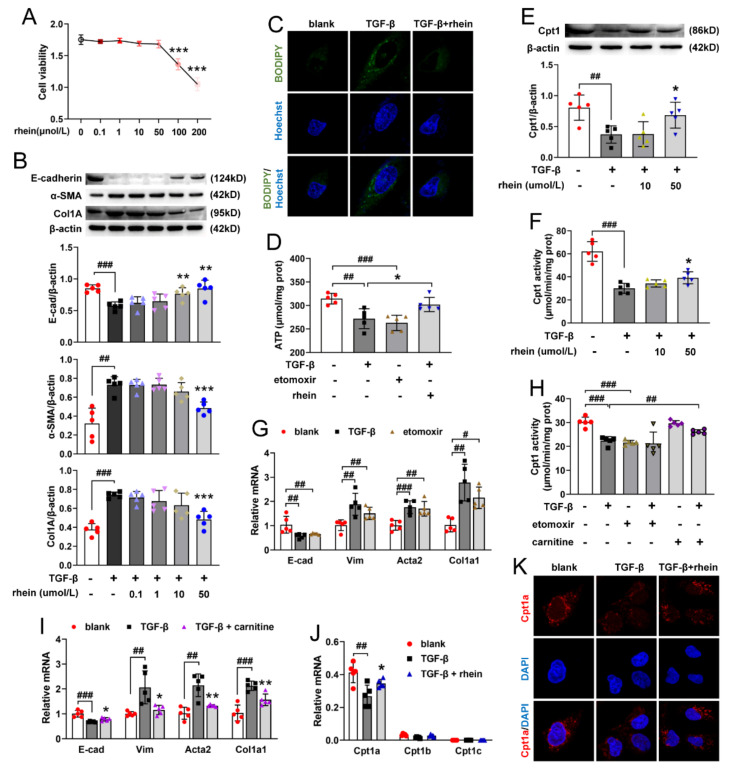
Rhein reverses TGF-β-induced EMT in RTE cells by promoting Cpt1a activity. (**A**) Cell viability in RTE cells incubated with rhein at indicated concentrations for 24 h. RTE cells were incubated with 10 ng/mL TGF-β and rhein at indicated concentrations for 24 h, (**B**) protein expressions of E-cadherin, α-SMA and Col1A; (**C**) representative cell images of lipid droplets stained with BODIPY (rhein at 50 μmol/L, ×630). (**D**) Intracellular ATP levels in RTE cells incubated with 10 ng/mL TGF-β and 50 μmol/L rhein, or with 40 μmol/L etomoxir, for 24 h. RTE cells were incubated with 10 ng/mL TGF-β and 10 or 50 μmol/L rhein for 24 h, (**E**) protein expression and (**F**) activity of Cpt1. RTE cells were incubated with 10 ng/mL TGF-β and 50 μmol/L rhein or 0.5 mmol/L carnitine, or with 40 μmol/L etomoxir, for 24 h, (**G**,**I**) mRNA expressions of E-cadherin, vimentin, Acta2 and Col1a1; (**H**) Cpt1 activity. RTE cells were incubated with 10 ng/mL TGF-β and 50 μmol/L for 24 h, (**J**) mRNA expressions of Cpt1a, Cpt1b and Cpt1c; (**K**) representative images of cells immunostained with Cpt1a (×630). Data were expressed as mean ± SD (*n* = 5). ^#^
*p* < 0.05, ^##^
*p* < 0.01, ^###^
*p* < 0.001, vs. blank cells; * *p* < 0.05, ** *p* < 0.01, *** *p* < 0.001, vs. TGF-β- or etomoxir-induced cells.

**Figure 4 molecules-27-02344-f004:**
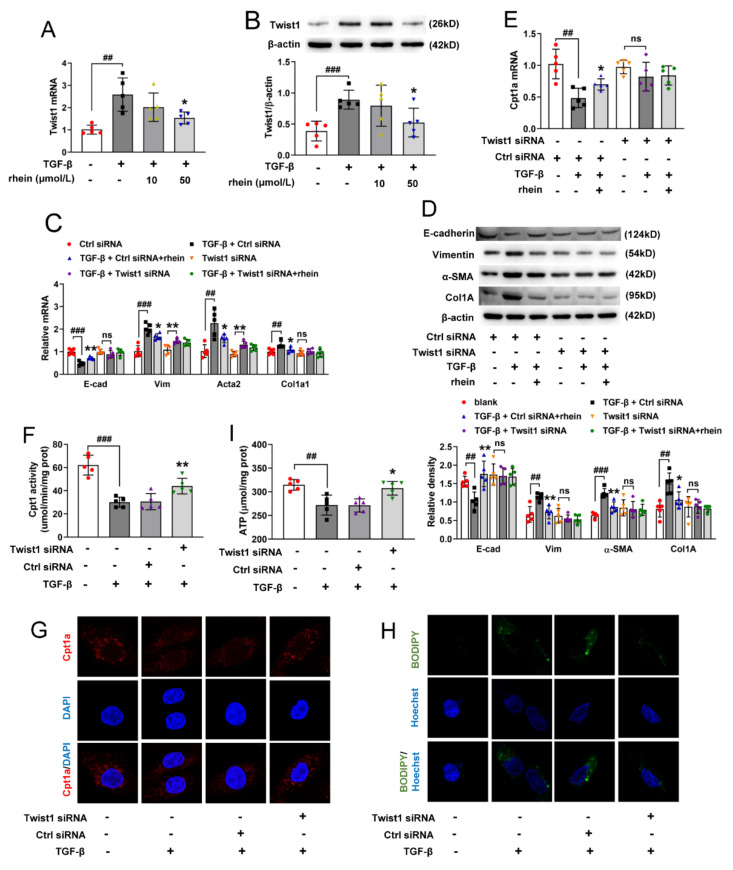
Twist1 is essential for Cpt1a-mediated FAO depression in RTE cells. (**A**) mRNA and (**B**) protein expression of Twist1 in RTE cells incubated with 10 ng/mL TGF-β and 10 or 50 μmol/L rhein for 24 h. After Twist1 transfection, RTE cells were incubated with 10 ng/mL TGF-β and 50 μmol/L rhein for 24 h, (**C**) mRNA and (**D**) protein expressions of E-cadherin, vimentin, Acta2 and Col1a1, (**E**) Cpt1a mRNA and (**F**) Cpt1 activity were detected; (**G**,**H**) representative images of cells immunostained with Cpt1a or stained with BODIPY (×630); (**I**) intracellular ATP levels. Data were expressed as mean ± SD (*n* = 5). ^##^
*p* < 0.01, ^###^
*p* < 0.001, vs. blank cells; * *p* < 0.05, ** *p* < 0.01, vs. TGF-β-induced cells.

**Figure 5 molecules-27-02344-f005:**
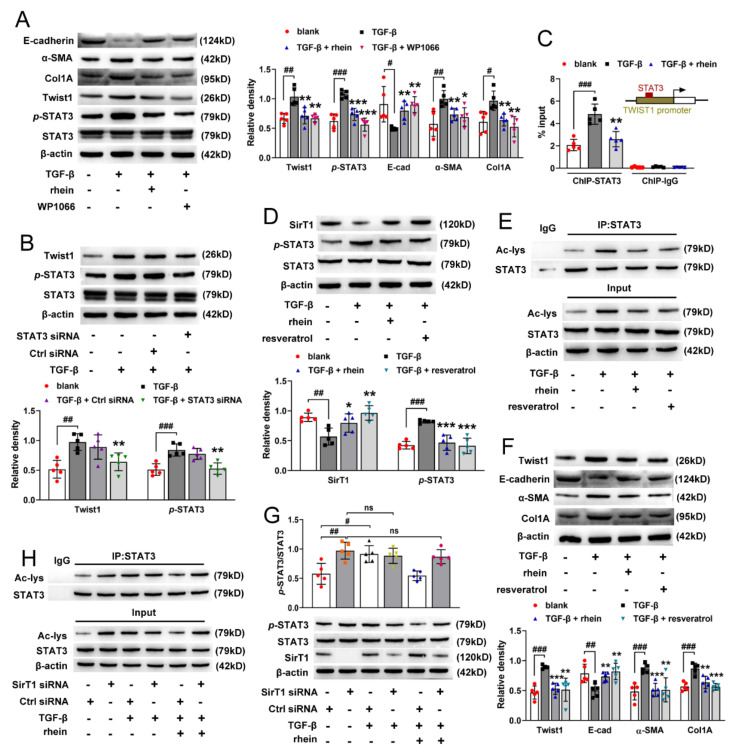
Rhein inhibits Twist1 expression by regulating SirT1/STAT3 pathway. (**A**) Protein expressions of Twist1, *p*-STAT3, E-cadherin, α-SMA and Col1A in RTE cells incubated with 10 ng/mL TGF-β and 50 μmol/L rhein or 2 μmol/L WP1066 for 24 h. (**B**) Protein expressions of Twist1 and *p*-STAT3 in RTE cells incubated with 10 ng/mL TGF-β after STAT3 transfection. (**C**) ChIP assay for STAT3 and Twist1 in RTE cells incubated with 10 ng/mL TGF-β and 50 μmol/L rhein for 24 h. RTE cells were incubated with 10 ng/mL TGF-β and 50 μmol/L rhein or 10 μmol/L resveratrol for 24 h, (**D**) protein expressions of SirT1 and *p*-STAT3; (**E**) protein expression of acetylated STAT3 (*n* = 4); (**F**) protein expressions of Twist1, E-cadherin, α-SMA and Col1A. After SirtT1 transfection, RTE cells were incubated with 10 ng/mL TGF-β and 50 μmol/L rhein for 24 h, (**G**) protein expressions of SirT1 and *p*-STAT3; (**H**) protein expression of acetylated STAT3 (*n* = 4). Data were expressed as mean ± SD (*n* = 5). ^#^
*p* < 0.05, ^##^
*p* < 0.01, ^###^
*p* < 0.001, vs. blank cells; * *p* < 0.05, ** *p* < 0.01, *** *p* < 0.001, vs. TGF-β-induced cells.

**Figure 6 molecules-27-02344-f006:**
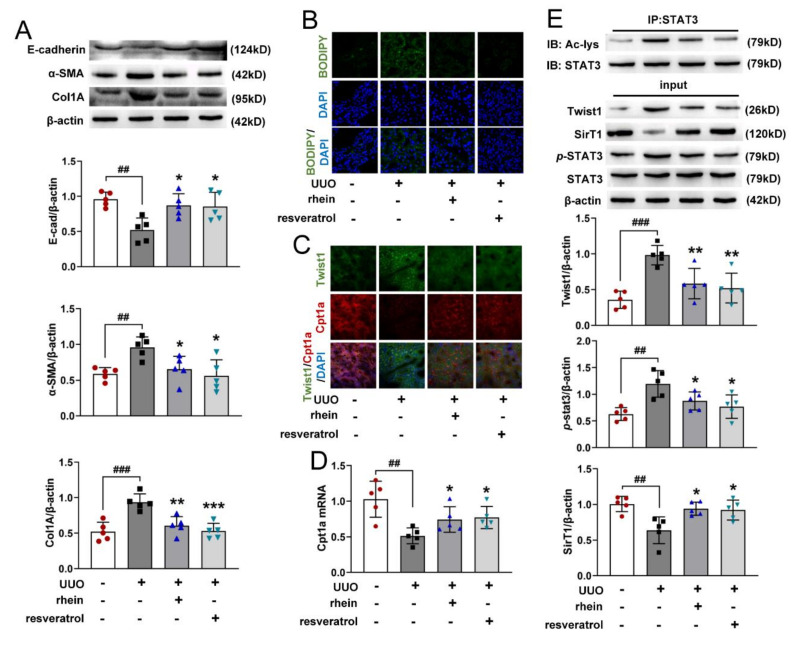
Rhein promotes Cpt1a-mediated FAO via SirT1/STAT3/Twist1 pathway in UUO-induced rats. Rat UUO operation was in the same manner as in Figure 1. The rats were i.g. administered with or without rhein at 100 mg/kg or resveratrol at 20 mg/kg per day from days 1 to 14, (**A**) protein expressions E-cadherin, α-SMA and Col1A of kidney samples; (**B**,**C**) representative images of kidney samples stained with BODIPY and immunostained with Cpt1a and Twist1 (×630, *n* = 3); (**D**) mRNA expression of Cpt1a; (**E**) protein expressions for Twist1, SirT1, *p*-STAT3 and acetylated STAT3. Data were expressed as mean ± SD (*n* = 5). ^##^ *p* < 0.01, ^###^ *p* < 0.001, vs. sham rats; * *p* < 0.05, ** *p* < 0.01, *** *p* < 0.001, vs. UUO rats.

**Figure 7 molecules-27-02344-f007:**
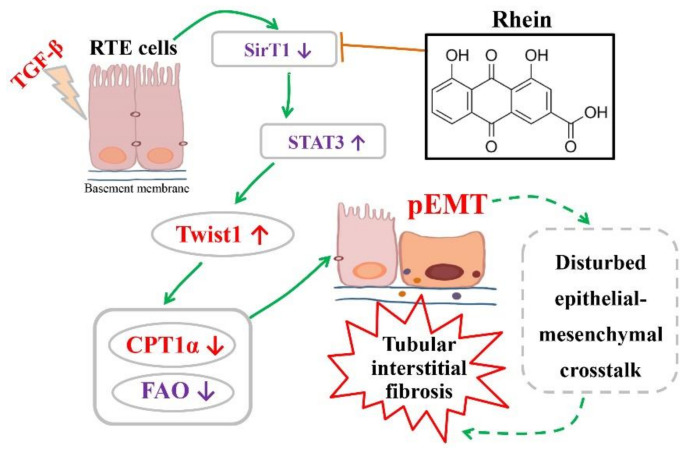
The proposed pathway of TGF-β-induced pEMT in RTE cells and the inhibitory effect of rhein.

## Data Availability

RNA-seq data displayed are deposited in NCBI Sequence Read Archive (BioProject ID: PRJNA785489). All other remaining data are available within the article and Appendix A, or available from the authors upon request.

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
