# Peer review of "Rhein Improves Renal Fibrosis by Restoring Cpt1a-Mediated Fatty Acid Oxidation through SirT1/STAT3/twist1 Pathway"

_molecules, 2022, doi:10.3390/molecules27072344_

Round 1
Reviewer 1 Report
Song et al paper shows that Rhein improves renal fibrosis by restoring Cpt1a-mediated fatty acid oxidation through SirT1/STAT3/Twist1 pathway, the experiments are well designed and all results support their hypothesis
Author Response
Song et al paper shows that Rhein improves renal fibrosis by restoring Cpt1a-mediated fatty acid oxidation through SirT1/STAT3/Twist1 pathway, the experiments are well designed and all results support their hypothesis.
Response 1: Thank you for your review and recognition.
Reviewer 2 Report
This is an interesting ms. on a potentially important topic. All parts of the ms. are well-done and sections are well-written. The paper might be of relevance.
Author Response
This is an interesting ms. on a potentially important topic. All parts of the ms. are well-done and sections are well-written. The paper might be of relevance.
Response 1: Thank you for your review and recognition.
Reviewer 3 Report
In this manuscript, authors investigated the anti-fibrotic mechanism for rhein in a rat model of renal fibrosis. This study employed UUO-driven renal fibrosis for in vivo studies followed by RNA-seq to identify novel pathway analysis activated by rhein and subsequent validation studies using in vitro TGF-beta-induced EMT studies using RTE cells. Overall, this is a comprehensive study to assess rhein anti-fibrotic mechanisms. However, as cited by authors (reference # 14), prior work already demonstrated that rhein inhibits stat3 phosphorylation and numerous studies by others demonstrated the importance of stat3 phosphorylation in TGF-beta-induced Twist1 and mesenchymal induction. Thus, data in in Figures 1, 4 and 5 is not new and the findings are redundant. Otherwise, rhein/Cpt1a/fatty acid oxidation (FOA) seems to be the ONLY but very interesting molecular axis identified from this study.
- In Figure 2A, Besides FAO-linked genes, RNA-seq also revealed cAMP pathway. Of note, a number of studies demonstrated cAMP potential to exert anti-fibrotic actions in lung and other tissue fibrosis models (eg. PMIDs: 31563998, 25622251). Surprisingly, authors didn’t perform experiments to evaluate other pathways identified in RNA-seq including rhein/Cpt1a/fatty acid-mediated cAMP elevation. While author’s data indicate that rhein/Cpt1a/FAO activation reverse EMT, authors should at least verify rhein capacity in cAMP induction. Also, it would it helpful to consider cAMP path-specific genes revealed from RNA-seq should be listed as supplemental.
- Among the pathways listed in Figure 2A, authors did not justify the reason for exploring the FAO alone in this disease model.
- In Figures 1 and 2, there is no rhein (alone)-treated group! This control is necessary to identify rhein-target genes. Throughout the figures, authors labelled the 3rd group as Rhein however, it should be corrected as UUP-> rhein or TGF-beta-> Lack of this rhein alone also limits RNA-seq utility.
Minor:
- Abstract was not presented well. It should be re-written to emphasize the key findings with relevant background.
- In 2.2, instead of saying UUO-induced renal fibrosis in rats “may be related to” FAO ..., authors specify a direction. Because RNA-seq revealed fatty acid oxidation genes, it can be stated as UUO-induced renal fibrosis in rats “is associated with” FAO …
Author Response
Point 1: In this manuscript, authors investigated the anti-fibrotic mechanism for rhein in a rat model of renal fibrosis. This study employed UUO-driven renal fibrosis for in vivo studies followed by RNA-seq to identify novel pathway analysis activated by rhein and subsequent validation studies using in vitro TGF-beta-induced EMT studies using RTE cells. Overall, this is a comprehensive study to assess rhein anti-fibrotic mechanisms. However, as cited by authors (reference # 14), prior work already demonstrated that rhein inhibits stat3 phosphorylation and numerous studies by others demonstrated the importance of stat3 phosphorylation in TGF-beta-induced Twist1 and mesenchymal induction. Thus, data in in Figures 1, 4 and 5 is not new and the findings are redundant. Otherwise, rhein/Cpt1a/fatty acid oxidation (FOA) seems to be the ONLY but very interesting molecular axis identified from this study.
Response 1: Thanks for your interesting review. We know that the data in Figures 1, 4 and 5 are not new data, but these results can maintain the integrity, logicality and readability of this manuscript, so they are not redundant.
Point 2: In Figure 2A, Besides FAO-linked genes, RNA-seq also revealed cAMP pathway. Of note, a number of studies demonstrated cAMP potential to exert anti-fibrotic actions in lung and other tissue fibrosis models (eg. PMIDs: 31563998, 25622251). Surprisingly, authors didn’t perform experiments to evaluate other pathways identified in RNA-seq including rhein/Cpt1a/fatty acid-mediated cAMP elevation. While author’s data indicate that rhein/Cpt1a/FAO activation reverse EMT, authors should at least verify rhein capacity in cAMP induction. Also, it would it helpful to consider cAMP path-specific genes revealed from RNA-seq should be listed as supplemental.
Response 2: Thanks for the valuable suggestion. We added the research progress of cAMP pathway on renal fibrosis and the effect of rhein on cAMP path-specific genes in the third paragraph of 2.2 (in red). And a heatmap of cAMP path-specific genes was also added in the supplementary material (as Fig. S1).
Point 3: Among the pathways listed in Figure 2A, authors did not justify the reason for exploring the FAO alone in this disease model.
Response 3: Thanks a lot. We are not exploring FAO alone, but the relationship between FAO and EMT. However, according to your suggestion, we explained the reason for the study (in red) at the beginning of the fourth paragraph of 2.2.
Point 4: In Figures 1 and 2, there is no rhein (alone)-treated group! This control is necessary to identify rhein-target genes. Throughout the figures, authors labelled the 3rd group as Rhein however, it should be corrected as UUO-> rhein or TGF-beta-> Lack of this rhein alone also limits RNA-seq utility.
Response 4: Thanks for your kind suggestion. However, this manuscript is not to study the target genes of rhein, but to study the recovery effect of rhein on the differential genes caused by UUO. Therefore, the rhein (alone)-treated group is not of great significance to this study. Thank you for your kind reminder, we have corrected the mark of rhein group in all figures according to your suggestion.
Minor:
Point 5: Abstract was not presented well. It should be re-written to emphasize the key findings with relevant background.
Response 5: Thanks, we have rewritten the abstract according to your valuable suggestion.
Point 6: In 2.2, instead of saying UUO-induced renal fibrosis in rats “may be related to” FAO ..., authors specify a direction. Because RNA-seq revealed fatty acid oxidation genes, it can be stated as UUO-induced renal fibrosis in rats “is associated with” FAO …
Response 6: Thanks very much, we have corrected the corresponding part in title 2.2 (indicated in red).
Please check all corrections in the attached manuscript.
Thank you very much!
